

# LC-TMNet: learned lossless medical image compression with tunable multi-scale network

Hengrui Liao and Yue Li

School of Computer, University of South China, HengYang, HuNan, China

## ABSTRACT

In medicine, high-quality images are crucial for accurate clinical diagnosis, making lossless compression essential to preserve image integrity. Neural networks, with their powerful probabilistic estimation capabilities, seamlessly integrate with entropy encoders to achieve lossless compression. Recent studies have demonstrated that this approach outperforms traditional compression algorithms. However, existing methods have yet to adequately address the issue of inaccurate probabilistic estimation by neural networks when processing edge or complex textured regions. This limitation leaves significant room for improvement in compression performance. To address these challenges, this study proposes a novel lossless image compression method that employs a flexible tree-structured image segmentation mechanism. Due to the close relationships between subimages, this mechanism allows neural networks to fully exploit the prior knowledge of encoded subimages, thereby improving the accuracy of probabilistic estimation in complex textured regions of unencoded subimages. In terms of network architecture, we have introduced an attention mechanism into the UNet network to enhance the accuracy of probabilistic estimation across the entire subimage regions. Additionally, the flexible tree-structured image segmentation mechanism enabled us to implement variable-speed compression. We provide benchmarks for both fast and slow compression modes. Experimental results indicate that the proposed method achieves state-of-the-art compression speed in the fast mode. In the slow mode, it attains state-of-the-art performance.

## INTRODUCTION

With the rapid advancement of imaging technology, medical images have become increasingly vital in clinical diagnosis. High image quality is critical, as even small details can impact diagnostic accuracy, necessitating the use of lossless compression for storage. In lossless compression, arithmetic encoders can achieve near-optimal results for a symbol $s$ and its probability $P$, requiring $-\log_2 P$ bits according to the source coding theorem (*Shannon, 1948*). Thus, compression algorithms using arithmetic coding must first establish a probabilistic model to estimate $P$. The more accurate this estimation, the fewer bits are needed, making efficient probabilistic modeling a key focus of research.

Corresponding author
Yue Li, liyue@usc.edu.cn

Learning-based image compression methods have made significant progress in improving image probability modeling performance. Typical methods, such as PixelRNN (*Van Den Oord, Kalchbrenner & Kavukcuoglu, 2016*), use autoregressive models to predict the conditional probability distribution of images pixel by pixel. In contrast, L3C (*Mentzer et al., 2019*) employs a multi-scale entropy model to estimate the probability of the entire image by leveraging the prior distribution in the latent space, effectively reducing the computational complexity associated with pixel-by-pixel predictions. However, these methods apply a uniform modeling strategy to all pixels, and their probability estimation by neural networks remains inaccurate when dealing with edge regions or complex textures.

Medical images typically contain complex anatomical structures and subtle lesion textures, which are crucial for medical diagnosis. However, these intricate texture structures also make medical images difficult to compress efficiently. Uniform modeling strategies often struggle with rapidly changing texture regions, failing to accurately estimate the probability distribution of the image. As a result, there is still considerable room for improvement in the compression of medical images. This study adopts a subimage partitioning scheme to enable a more flexible modeling strategy. By utilizing the strong correlations between subimages to provide prior information, the inference capability of the neural network is significantly enhanced. In the probability estimation module, we integrate attention mechanisms with a UNet structure to further improve the accuracy of the model's probability estimation.

Specifically, we propose an enhanced probability estimation method based on flexible subimage prior knowledge, aimed at improving estimation accuracy in these regions. Drawing on the subimage partitioning schemes of High Efficiency Video Coding (*Sze, Budagavi & Sullivan, 2014*) and Versatile Video Coding (*Bross et al., 2021*), we propose the tree-structured flexible subimage partitioning scheme illustrated in Fig. 1. Initially, a quadtree is used for the first-Level partitioning (Level 1 subimage) of the image. Subsequently, a binary tree is employed for further partitioning of the first-level subimages (Level 2 subimage), specifically splitting subimages B, C, and D according to odd and even columns. Since odd and even columns are spatially adjacent, this ensures that the $subimage_{left}$ can provide strong prior knowledge for the $subimage_{right}$, i.e., $P(subimage_{right}|subimage_{left})$. With this strong prior, the neural network can achieve precise probabilistic inference for the $subimage_{right}$, including complex textured regions. In terms of network design, we incorporated the Squeeze-and-Excitation (SE) attention mechanism (*Hu, Shen & Sun, 2018*) into the probability estimation module. This method enhances model performance by recalibrating the feature responses in the convolutional network channels. It significantly improves the network's sensitivity to useful information while suppressing the interference of irrelevant information, thereby enhancing the model's probability estimation capability.

Based on the flexible subimage partitioning scheme, we propose a variable-speed compression framework. As shown in Fig. 2, subimage A is first compressed using a traditional encoder. Subsequently, the compression process can follow two paths: the orange High Way and the green Low Way. The orange High Way targets compression

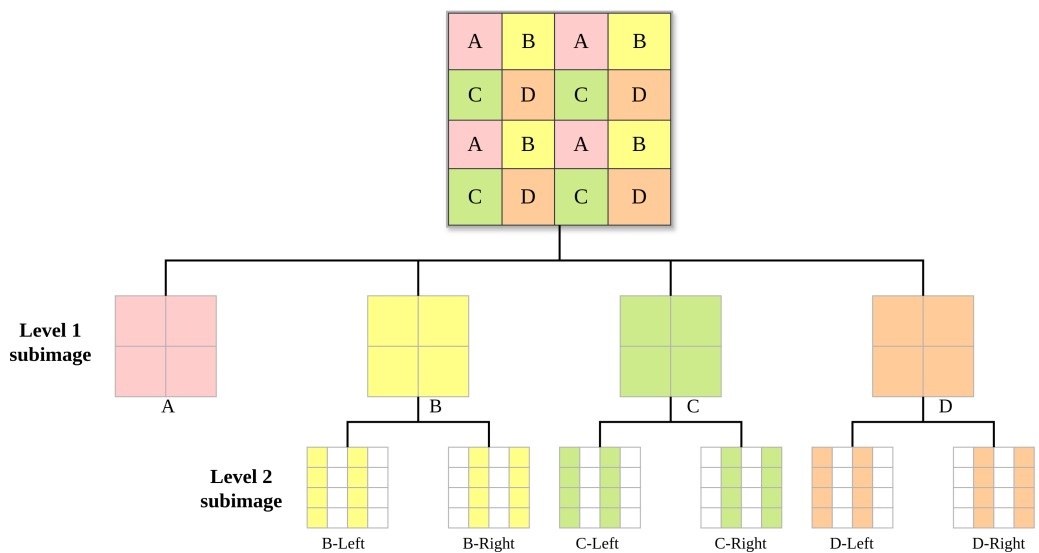

**Figure 1** **Flexible subimage partitioning scheme with a tree structure.** The original image is partitioned into Level 1 subimages using a quadtree. The Level 1 subimages are then partitioned into Level 2 subimages using a binary tree.

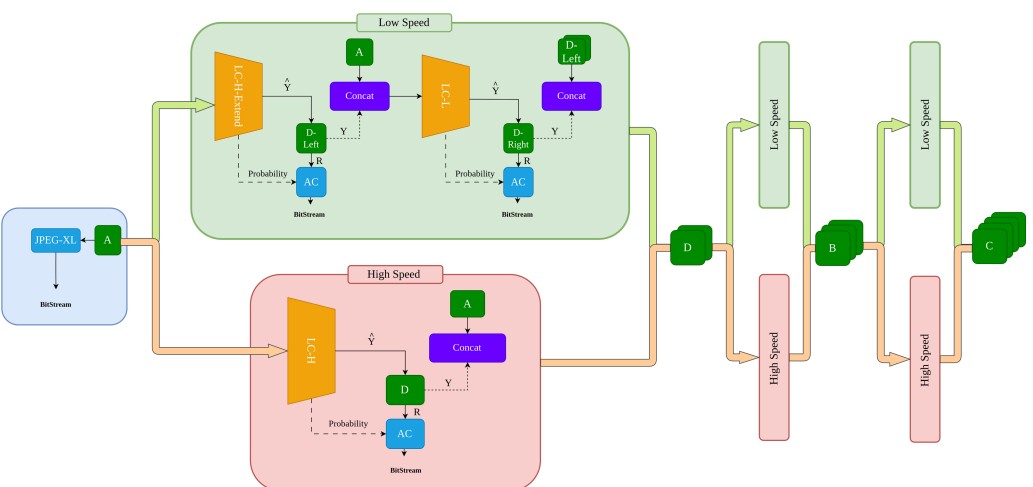

**Figure 2** **Overview of the compression process.** The compression process comprises a rapid orange channel and a slower green channel. By flexibly integrating the orange and green channels, Tunable Multi-Scale compression is realized.

speed, while the green Low Way focuses on compression efficiency. This will be detailed in the ''Method'' section. Thus, this compression framework allows for arbitrary combinations of compression routes. In our experiments, we provide two benchmark tests: one using the orange fast compression path $a \rightarrow d \rightarrow b \rightarrow c$ and the other using the green slow compression path $a \rightarrow d_{left} \rightarrow d_{right} \rightarrow b_{left} \rightarrow b_{right} \rightarrow c_{left} \rightarrow c_{right}$.

The main contributions are summarized as follows:

- Subimage partitioning scheme: By employing a flexible subimage partitioning scheme, the neural network effectively leverages subimage information, addressing the issue of inaccurate probability estimation in regions with complex textures.
- Network architecture improvement: An attention mechanism was incorporated into the probabilistic estimation model, boosting the model's probabilistic estimation capability. To the best of our knowledge, this is the first application of an attention mechanism in the field of lossless compression, thereby extending its application scope.
- Flexible compression process combination: The proposed compression framework allows for flexible combinations of compression processes. In the fast compression scheme, we achieved the highest compression speed; in the slow compression scheme, we attained state-of-the-art compression performance.

# RELATED WORK

## Likelihood-based generative models

Arithmetic coding requires obtaining the probability of each symbol; hence, early lossless compression commonly employed autoregressive models. In such models, each pixel is compressed based on previously encoded pixels. For instance, PixelRNN (*Van Den Oord, Kalchbrenner & Kavukcuoglu, 2016*) processes image pixels sequentially using a recurrent neural network (RNN), while PixelCNN (*Van den Oord et al., 2016*) models pixel values using a masked convolutional neural network (CNN). Both methods predict pixels based on the conditional distribution of all preceding pixels, modeling pixels as the product of conditional distributions $p(x) = \prod p(x_i | x_1, \ldots, x_{i-1})$, where $xi$ represents a single pixel.

PixelCNN++ (*Salimans et al., 2017*), an improved version of PixelCNN, reduces dependence on previous channels by modeling the joint distribution of each pixel. Additionally, PixelCNN++ introduces discrete logistic mixture likelihood, multi-scale downsampling, and extra shortcut connections, which enhance compression performance and reduce processing time. Despite these improvements, PixelCNN++ retains the inherent limitation of autoregressive models, requiring network computation for each pixel, resulting in a time complexity of $O(W \times H)$, where $W$ is the width and $H$ is the height, leading to prolonged inference times.

## Multi-scale entropy models

To optimize compression efficiency and address time constraints. L3C has made pioneering contributions in this field. L3C uses a hierarchical model to predict all pixels of an entire image. Specifically, L3C considers the image $X$ itself as the zeroth layer and utilizes the first latent space $Z_1$ to construct a probability model $P(X|Z_1)$. Similarly, compressing $Z_1$ requires the second latent space $Z_2$, with the probability expression $P(Z_1|Z_2)$, and so on. However, since the last layer of features $Z_3$ does not have a subsequent layer of features to serve as a prior, $P(Z_3)$ is set to a fixed value. This fixed value usually deviates significantly from the actual probability, resulting in poor compression performance for the last layer of features $Z_3$.

To address the lack of prior knowledge, residual compressor (RC) (*Mentzer, Van Gool & Tschannen, 2020*) and deep lossy plus residual (DLPR) (*Bai et al., 2024*) approach the

**Table 1 Method analysis and improvements.** This table highlights the key methods, including LC-FNet, PixelRNN, and RC, along with their advantages and disadvantages, while also demonstrating how our approach addresses the limitations of these methods.

| Method | Advantages | Disadvantages | Improvements |
|---|---|---|---|
| PixelRNN | Pixel-wise modeling effectively leverages contextual information, achieving high-precision lossless compression. | High computational complexity, leading to significant inference time costs. | Uses subimage level processing units to reduce complexity and improve efficiency. |
| RC | Convert the original image to a residual image, leverage the concentration of residual values to improve probability prediction accuracy. | Requires additional storage for lossy images and residual images; lossy components contribute limited information to residual prediction. | Compresses the first subimage losslessly, reducing storage overhead while enhancing residual prediction accuracy. |
| LC-FNet | By classifying pixels based on their frequency characteristics, the introduction of the pixel classification concept leads to improved compression efficiency. | Low and high frequency pixels occupy different regions with weak correlation, requiring a decomposition network for classification, which increases computational overhead. | Binary tree decomposition method that avoids extra computation and leverages the tight spatial relationship between even and odd pixels, providing strong priors. |

problem from the perspective of lossy compression. They use a lossy image as prior knowledge to predict the residual probability of the image, with the probability model $P(X_{residual}|X_{lossy})$. By combining the residual with the lossy image, they achieve lossless compression. These experiments provide crucial insights: a low bit-rate lossy image offers prior knowledge, transforming the original image into a residual image. Leveraging the concentrated nature of residuals improves the accuracy of probability prediction $P$.

In LC-FNet (*Rhee et al., 2022*), the authors deeply investigated residual characteristics from a frequency domain perspective and proposed a frequency decomposition network. This approach assumes that residual probability estimation in low-frequency regions is relatively straightforward, whereas high-frequency regions are more challenging. Thus, LC-FNet first compresses the low-frequency regions before processing the high-frequency regions, leveraging prior information from the low-frequency regions to improve the probability estimation of high-frequency regions. The probabilistic model is formulated as $P(X_{high-frequency}|X_{low-frequency})$. *Wang et al. (2023b)* proposed a novel decomposition approach based on image storage bytes, contrasting LC-FNet's frequency method. They split the image into the most significant byte (MSB) subimage for higher-order bits and the least significant byte (LSB) subimage for lower-order bits. The MSB, requiring lower compression bitrate, is encoded traditionally and used as input to a neural network to predict and encode the LSB.

In this paper, we are primarily inspired by LC-FNet, PixelRNN, and RC. While these methods show distinct advantages in specific scenarios, they still face limitations regarding computational complexity, compression efficiency, and flexibility. Building on these insights, this paper proposes a novel multi-scale compression framework that introduces subimage-level lossless priors and binary-tree decomposition strategy to overcome some of the bottlenecks of existing approaches. The specific improvements are summarized in Table 1.

# METHOD

The overall process of our method is shown in Fig. 2. First, subimage A is losslessly compressed using an advanced traditional compression algorithm. JPEG-XL (*Alakuijala et al., 2019*) is chosen because when the neural network lacks prior knowledge, it cannot perform effective probability estimation. In such cases, traditional algorithms demonstrate higher competitiveness. Subsequently, the compression process can follow either the high-speed orange channel or the slow-speed green channel.

In the orange channel, subimage A is input into the learning-based compressor LC-H, which outputs the predicted pixel $\hat{Y}$ and residual probability $P$ for subimage D. The predicted pixels $\hat{Y}$ are subtracted from the true pixels $Y$ of subimage D to obtain the residual $R$. $R$ and $P$ are then input into the arithmetic coding (AC) for lossless compression.

In the green channel, subimage A is input into the learning-based compressor LC-H-Extend, which outputs the predicted pixels $\hat{Y}$ and residual probability $P$ for subimage $D_{left}$. The predicted pixels $\hat{Y}$ are subtracted from the true pixels $Y$ of subimage $D_{left}$ to obtain the residual $R$. $R$ and $P$ are then input into the arithmetic coding for lossless compression. The encoded $A$ and $D_{left}$ are combined as prior information and input into LC-L. The process is similar to LC-H, resulting in the residuals $R$ and residual probability $P$ for subimage $D_{right}$, which are then input into the AC for compression. Since $cat(D_{left}, D_{right}) \rightarrow D$, the outputs of the orange and green channels are the same. The compression processes for subimages B and C follow the same logic as for subimage D.

Algorithm 1 describes the entire compression workflow of Fig. 2 in pseudocode form. In line 1, the pseudocode outlines the decomposition of the input image $I$ into four level-1 subimages: $I_a$, $I_b$, $I_c$, and $I_d$, corresponding to the process illustrated in Fig. 1. Lines 2 to 4 describe the process of compressing subimage $I_a$ using JPEG-XL, as shown in Fig. 2, ultimately generating the initial bitstream $bitStream_{I_a}$.

Next, the pseudocode details the process in the orange channel (high-speed channel) shown in Fig. 2. In lines 6 to 8, the algorithm first retrieves the real subimage *realimg* (denoted as $Y$ in Fig. 2) at the current position *pos*. The LC-H module is then used to obtain the predicted subimage *predimg* (denoted as $\hat{Y}$ in Fig. 2) and the residual probability *resprob* (denoted as *Probability* in Fig. 2). By subtracting the predicted subimage *predimg* from the real subimage *realimg*, the residual subimage *resimg* (denoted as $R$ in Fig. 2) is obtained. Line 20 describes the process of feeding the residual subimage *resimg* and residual probability *resprob* into the entropy encoder for compression.

Subsequently, the pseudocode describes the process in the green channel (slow-speed channel) as illustrated in Fig. 2. Lines 10 to 11 explain the LC-H-Extend compression process within the green channel. The LC-H-Extend module extracts the odd rows from the residual subimage *resimg* and residual probability *resprob* to generate $resimg_{left}$ and $resprob_{left}$, respectively, which are then input into the entropy encoder for compression. Lines 13 to 16 describe the LC-L compression process within the green channel, as also depicted in Fig. 2. First, the real subimage $realimg_{left}$ is concatenated with *inputs* to update *inputs*. The LC-L module is then used to obtain the predicted subimage $predimg_{right}$ and the residual probability resprobright for the even columns at the current position. The residual

subimage $resimg_{right}$ is obtained by subtracting $predimg_{right}$ from $realimg_{right}$, and both $resimg_{right}$ and $resprob_{right}$ are subsequently fed into the entropy encoder for compression.

---

**Algorithm 1** Compress Algorithm

---

**Require:** LC-TMnet *model*; Awaiting compressed image $I$;

**Ensure:** Compressed bitstream *bitStream*

1: Decompose image $I$ into four level 1 subimages: $I_a$, $I_b$, $I_c$, $I_d$
2: Compress subimage $I_a$ using JPEG XL to obtain the bitstream $bitStream_{I_a}$
3: $bitstream \leftarrow bitstream_{I_a}$
4: $inputs \leftarrow I_a$
5: **for** *pos* in [d, b, c] **do**
6:      $realimg \leftarrow getrealimg(pos, I_b, I_c, I_d)$
7:      $predimg, resprob \leftarrow model(inputs, \text{lc-h})$
8:      $resimg \leftarrow realimg - predimg$
9:      **if** use slow channel **then**
10:          $(resimg_{left}, resprob_{left}) \leftarrow model(resimg, resprob, \text{lc-h-extend})$
11:          $bitstream \leftarrow bitstream + entropyEncoder(resimg_{left}, resprob_{left})$
12:          /*=====LC-H-Extend compression completed=====*/
13:          $inputs \leftarrow concat(inputs, realimg_{left})$
14:          $predimg_{right}, resprob_{right} \leftarrow model(inputs, \text{lc-l})$
15:          $resimg_{right} \leftarrow realimg_{right} - predimg_{right}$
16:          $bitstream \leftarrow bitstream + entropyEncoder(resimg_{right}, resprob_{right})$
17:          /*=====LC-L compression completed=====*/
18:          $inputs \leftarrow concat(inputs, realimg_{right})$
19:      **else**
20:          $bitStream \leftarrow bitstream + entropyEncoder(resimg, resprob)$
21:          /*=====LC-H compression completed=====*/
22:          $inputs \leftarrow concat(inputs, realimg)$
23:      **end if**
24: **end for**
25: **return** $bitStream$

---

## LC-H architecture

The Fig. 3 shows the LC-H architecture. We will detail it in three parts: Preprocessing, Residual Image, and Residual Probability Distribution.

*Preprocessing.* Three ResBlocks are used to extract features from the input Level 1 subimage $X_{in} \in R^{N \times \frac{W}{2} \times \frac{H}{2}}$, where $N$ represents the number of previously compressed Level 1 subimages, with values ranging $N \in \{1, 2, 3\}$. in the illustration, $N$ is 3. $\frac{W}{2}$ represents the width of the Level 1 subimage, $\frac{H}{2}$ represents the height of the Level 1 subimage. The extracted features are transformed into a matrix representation called the "latent space", which will be used as input for subsequent probability prediction and image prediction.

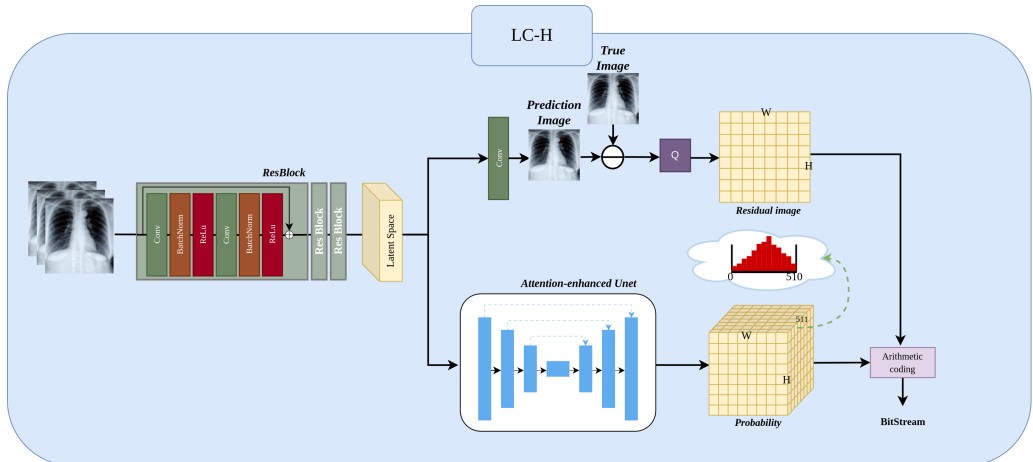

**Figure 3** **Implementation of LC-H architectures.** LC-H is designed to obtain the residual values and residual probability for Level 1 subimages.

In determining the preprocessing operations, we adopted the design principles from the classic ResNet-18 architecture. As demonstrated by *He et al. (2016)*, the use of ResBlocks has proven to be highly effective in feature extraction. Consequently, the preprocessing operations and parameter choices in this study are based on well-established best practices within the field. While parameter tuning is indeed an important task that could further enhance network performance, it is a distinct research area that falls beyond the current scope of this paper and is therefore not explored here.

*Residual Image.* The latent space, after passing through a convolutional layer, results in the predicted image $X_{pred} \in R^{1 \times \frac{W}{2} \times \frac{H}{2}}$, where 1 represents the predicted next Level 1 subimage. The residual image $X_{residual} \in R^{1 \times \frac{W}{2} \times \frac{H}{2}}$ is obtained by subtracting the predicted image from the true image. Since the predicted image is a floating point number in the range 0–255 and the true image is an integer in the range 0–255, the residual range is from −255 to 255 as a floating point number. Because entropy coding cannot directly compress negative and floating point numbers, we need to add 255 to the residual and then perform a quantization operation Q (rounding to the nearest integer). The final residual image $X_{residual}$ is thus an integer in the range 0 to 510.

*Residual probability distribution.* The latent space obtains the probability $P_{residual} \in R^{511 \times \frac{W}{2} \times \frac{H}{2}}$ through an attention-enhanced UNet. Here, The depth of 511 in the probability matrix arises because each residual pixel can take on an integer value between 0 and 510, resulting in 511 possible values. Consequently, the probability matrix has a depth of 511, representing the probability distribution for each possible value. The sum of these 511 probability equals 1. Finally, the probability distribution and the residual subimage are encoded using an entropy encoder.

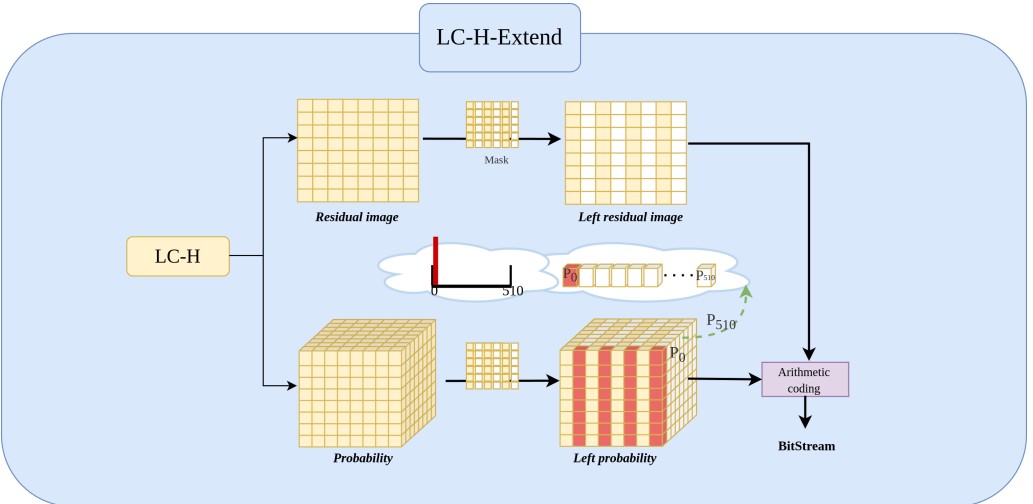

**Figure 4  Implementation of LC-H-Extend architectures.** LC-H-Extend is designed to obtain the residual values and residual probability for Level 2 left subimages.

### LC-H-Extend architecture

The Fig. 4 shows the LC-H architecture. The The LC-H-Extend module is an extension of the LC-H architecture, designed to retain the even columns of $X_{residual} \in R^{1 \times \frac{W}{2} \times \frac{H}{2}}$ while masking the odd columns. Specifically, the module uses a $Mask \in R^{1 \times \frac{W}{2} \times \frac{H}{2}}$ to keep the even columns of $X_{residual}$ unchanged and set the odd columns to a fixed value 0. The mask is given by Eq. (1):

$$Mask = \begin{cases} 1 & \text{if } W \text{ is even} \\ 0 & \text{if } W \text{ is odd} \end{cases}. \tag{1}$$

By applying $X_{residual} \cdot Mask = X_{left-residual}$, where all odd columns are 0. Consequently, experiment set the probability at the 0th position (out of 511 positions) in the odd columns of the probability matrix $P_{residual} \in R^{511 \times \frac{W}{2} \times \frac{H}{2}}$ to 1, indicating that the probability of odd column values being zero is 100%. Simultaneously, positions 1 to 510 in the odd columns are set to 0, indicating the probability of odd column values being non-zero is 0%. This yields the new probability matrix $P_{left-residual}$, as defined in Eq. (2).

$$P_{left-residual} = \begin{cases} P_{residual} & \text{if } W \text{ is even} \\ 1(100\%) & \text{if } W \text{ is odd and at position 0} \\ 0(0\%) & \text{if } W \text{ is odd and at positions 1 to 510} \end{cases}. \tag{2}$$

### LC-L architecture

Figure 5 shows the LC-L architecture. We will detail it in three parts: Preprocessing, Residual Image, and Residual Probability Distribution.

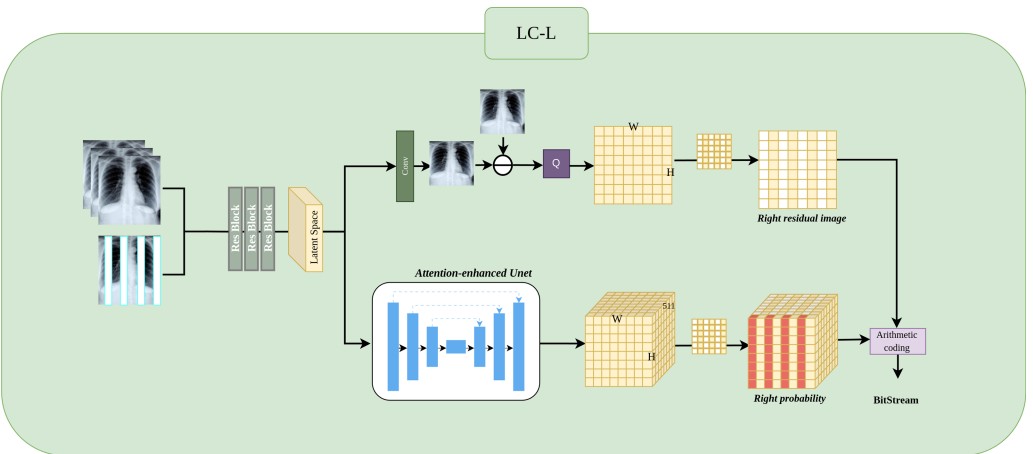

**Figure 5  Implementation of LC-L architectures.** LC-L is designed to obtain the residual values and residual probability for Level 2 right subimages.

*Preprocessing.*  The input Level 1 subimages $X_{in} \in R^{N \times \frac{W}{2} \times \frac{H}{2}}$ is concatenated with the Level 2 left subimage $X \in R^{1 \times \frac{W}{2} \times \frac{H}{2}}$, resulting in a new input $X_{in} \in R^{N+1 \times \frac{W}{2} \times \frac{H}{2}}$ where $N$ represents the number of previously compressed Level 1 subimages, with $N \in \{1, 2, 3\}$, and 1 indicates the previously compressed Level 2 left subimage. Next, feature extraction is performed through three ResBlocks, producing the feature representation as a latent space matrix.

*Residual image.*  This process is similar to LC-H and LC-H-Extend, where the residual $X_{residual}$ is obtained by subtracting the predicted image from the ground truth image, followed by applying a mask to obtain $X_{right-residual}$. The input Level 2 left subimage provides strongly correlated information for odd-column predictions, making the Level 2 right subimage predictions closer to the actual Level 2 right subimage. Consequently, the values of $X_{right-residual}$ tend to be concentrated around 255, which facilitates probability estimation.

*Residual probability distribution.*  This process is similar to LC-H and LC-H-Extend. The probability $P_{residual}$ is obtained through an attention-enhanced UNet. Since the Level 2 left subimage provides highly correlated experience for odd column prediction, $P_{residual}$ achieves more accurate probability estimation for the odd columns. Subsequently, we use a mask to extract the odd column probability from $P_{residual}$. Specifically, to mask the even columns of $P_{residual}$, we set the 0th position of the even columns to 1, while positions 1 to 510 in the even columns are set to 0. Ultimately, we obtain $P_{right-residual}$, which retains only the odd column probability.

## Attention-enhanced UNet architecture

As shown in Fig. 6, the UNet architecture (*Ronneberger, Fischer & Brox, 2015*) is enhanced with SE attention (*Hu, Shen & Sun, 2018*) to improve sensitivity to key information and probability estimation. After four downsampling and upsampling stages, the Softmax function ensures the predicted probabilities sum to 1.

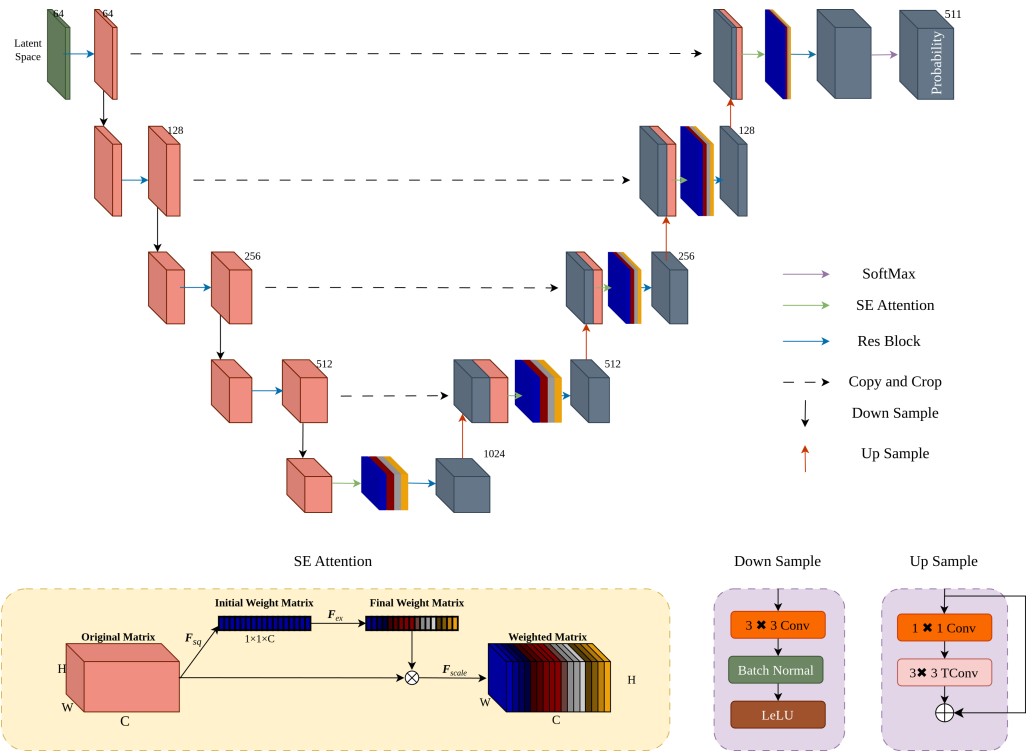

**Figure 6  Implementation of attention-enhanced UNet architecture.** To enhance the probability estimation capability of UNet, we integrated the SE attention mechanism into the UNet upsampling process.

During the upsampling process, we introduce the SE attention mechanism. Firstly, apply global average pooling $F_{sq}$ to the original matrix $X \in R^{C \times W \times H}$ to obtain the initial weight matrix $Z \in R^{C \times 1 \times 1}$. The mathematical expression for $F_{sq}$ is given by Eq. (3)

$$Z_c = \frac{1}{W \times H} \sum_{i=1}^{W} \sum_{j=1}^{H} X_{c,i,j} \tag{3}$$

where $Z_c$ is the channel-wise average of the original matrix $X$. Next, the initial weight matrix $Z$ is transformed into the final weight matrix $S$ through $F_{ex}$, which is a fully connected layer. The mathematical expression for $F_{ex}$ is provided in Eq. (4).

$$S = F_{ex}(Z) = \sigma(W_2 \cdot \text{ReLU}(W_1 \cdot Z)) \tag{4}$$

where $W_1$ and $W_2$ are weight matrices of the fully connected layers, ReLU denotes the ReLU activation function, and $\sigma$ denotes the sigmoid activation function. Finally, the final weight matrix $S$ is used in the scaling operation $F_{scale}$, where it is multiplied channel-wise with the original matrix $X$ to obtain the weighted matrix $\hat{X}$. The mathematical formulation for $F_{scale}$ is shown in Eq. (5).

$$\hat{X}_{c,i,j} = F_{scale}(X,S) = S_c \cdot X_{c,i,j} \tag{5}$$

where $\hat{X}$ is the weighted matrix obtained by scaling the original matrix $X$ with the final weight vector $S$.

## Loss function

The loss functions for LC-H and LC-L include the subimage prediction loss (SPL) and the probability prediction loss (PPL). LC-H-Extend does not include any loss functions, as it is an extension of LC-H and does not require separate loss functions.

*Subimage prediction loss.* We define the subimage prediction loss as the absolute difference between the predicted pixel values and the original pixel values, which is known as the mean absolute error loss in loss function terminology. It is mathematically defined in Eq. (6):

$$L_{SPL}(y,\hat{y}) = \frac{1}{N}\sum_{i=1}^{N}\left|y_i - \hat{y}_i\right| \tag{6}$$

where $N$ represents the total number of pixels in the image in LC-H, $N$ represents the number of pixels in the odd columns of the image in LC-L, $y$ represents the original pixel values, and $\hat{y}$ represents the predicted pixel values by the neural network.

*Probability prediction loss.* For the probability prediction loss function, cross-entropy loss was selected due to its well-established effectiveness in classification tasks, as demonstrated in various studies (*Mentzer et al., 2019*; *Rhee et al., 2022*; *Wang et al., 2023a*). The cross-entropy loss function (*De Boer et al., 2005*) is used to minimize the divergence between the predicted probability distribution and the true label distribution, thereby aligning the model's output distribution as closely as possible with the actual distribution. It is mathematically defined in Eq. (7):

$$L_{PPL}(p,\hat{p}) = -\frac{1}{N}\sum_{i=1}^{N}\sum_{j=1}^{C}p_{i,j}\cdot\log(\hat{p}_{i,j}) \tag{7}$$

where $N$ represents the total number of pixels in the image in LC-H, $N$ represents the number of pixels in the odd columns of the image in LC-L, and $C$ represents the number of residual value classes, ranging from 0 to 510, making a total of 511 classes. $p_{i,j}$ denotes the probability that the $i$-th pixel actually belongs to the $j$-th class. More precisely, if the $i$-th pixel belongs to the $j$-th class, $p_{i,j}$ equals 1; otherwise, it is 0. $\hat{p}_{i,j}$ is the model's predicted probability that the $i$-th pixel belongs to the $j$-th class.

## EXPERIMENT AND ANALYSIS

In this section, we first introduce the experimental setup, followed by a comparative analysis of the proposed method with the current state-of-the-art lossless compression techniques. Subsequently, we use probability heatmaps to visually demonstrate the effectiveness of the proposed design in improving probability estimation accuracy in complex texture regions. Finally, we validate the proposed design through ablation experiments, showing its effectiveness in addressing the impact of complex textures on compression performance.

## Experimental setup

*Datasets.* To comprehensively validate the effectiveness of the proposed method in practical applications, the experiments selected medical imaging datasets of different organs and grayscale processed natural image datasets as the test sets.

The datasets were chosen to ensure comprehensive coverage of common imaging modalities, thereby enhancing the model's generalizability. These datasets were sourced from reputable open-source platforms such as Kaggle, which not only facilitates the reproducibility of our experiments but also ensures that the data usage complies with legal and ethical standards. Moreover, these datasets have been widely used in previous research, underscoring their reliability and broad acceptance within the academic community.

It should be noted that there are significant differences between medical images (such as CT and MRI) and natural images in terms of the number of channels and image content. Medical images are typically single-channel grayscale images. Therefore, we converted natural images to grayscale to simulate the single-channel characteristics of medical images. This conversion ensures that the processed grayscale natural images are similar to medical images in terms of channel structure while retaining rich texture information.

(1) *Medical Imaging Dataset:* The dataset includes images from four different medical imaging datasets: the Large COVID-19 CT scan slice dataset (*Maftouni et al., 2021*), the COVID-19 Radiography Dataset (*Viradiya, 2021*), the Brain Tumor MRI Images dataset (*Bhuvaji et al., 2020*), and the Breast Ultrasound Images Dataset (*Al-Dhabyani et al., 2020*). In this experiment, 100 images were randomly selected from each dataset to serve as the test set.

(2) *COCO:* COCO dataset (*Lin et al., 2014*) is a large natural image dataset, and its complexity can thoroughly validate our model's performance in handling complex images. To maintain consistency with the characteristics of medical images, we converted the images in the COCO dataset to grayscale. This preprocessing step aims to ensure the effectiveness of our method when processing grayscale medical images while also demonstrating our model's robustness on complex images. We randomly selected 200 images from this dataset to serve as the test set.

*Training.* To ensure data diversity and the generalization ability of the model, we employed the Flickr2k dataset (*Lim et al., 2017*) for training our network, which contains 2,650 high-quality images. During the training process, we referred to the experimental configurations from *Tissier et al. (2023)* and *Rhee et al. (2022)*, and implemented a series of measures to increase the variability of the dataset. Specifically, we randomly extracted $128 \times 128$ pixel image patches from the original images. Throughout the training, we utilized the Adam optimizer (*Kingma & Ba, 2014*) with a batch size of 24. The training schedule lasted for 2,000 epochs, ensuring that the network had sufficient opportunity to learn complex patterns in the data. To achieve an optimal balance between exploration and exploitation, we set the initial learning rate to $1 \times 10^{-3}$. Additionally, to prevent overfitting, we implemented a learning rate decay strategy, halving the learning rate every 500 epochs.

**Table 2  Summary of compression performance comparison.** Performance is measured in Bits Per Pixel (bpp), the best performance is highlighted in bold, the second best performance is indicated with an asterisk (*).

| Method | Breast ultrasound images | Brain tumor MRI images | COVID-19 radiography | COVID-19 CT scan | COCO |
|---|---|---|---|---|---|
| PNG | 3.45 +30.7% | 3.52+29.4% | 3.10 +35.4% | 4.56 +10.7% | 4.31+26.4% |
| JPEG2000 | 3.01 +14.6% | 3.34 +22.8% | 2.77 +21.0% | 4.47 +8.5% | 4.10 +20.2% |
| WebP | 3.08 +16.7% | 3.27 +20.2% | 2.79 +21.8% | 4.35 +5.6% | 4.11 +20.5% |
| JPEG-LS | 2.98 +12.9% | 3.19 +17.3% | 2.66 +16.2% | 4.38 +6.3% | 4.03 +18.2% |
| FLIF | 2.86 +8.3% | 3.15 +15.8% | 2.59 +13.1% | 4.27 +3.6% | 3.95 +15.8% |
| JPEG-XL | 2.85 +8.0% | 3.13 +15.1% | 2.42 +5.7% | 4.34 +5.3% | 3.79 +11.1% |
| L3C | 3.21 +21.6% | 3.38 +24.3% | 3.05 +33.2% | 5.11 +24.0% | 4.09 +19.9% |
| L-Infinite | 3.09 +17.0% | 3.20 +17.6% | 2.73 +19.2% | 4.80 +16.5% | 3.89 +14.1% |
| LC-FNet | 2.78 +5.3% | 2.91 +7.0% | 2.40 +4.8% | 4.27 +3.6% | 3.71 +8.8% |
| LC-FNet++ | 2.83 +7.2% | 2.87 +5.5% | 2.44 +6.6% | 4.24 +2.9% | 3.70 +8.5% |
| DLPR | 2.76 +4.5% | 2.88 +5.9% | 2.34* +2.2% | 4.21 +2.2% | 3.68 +7.9% |
| Propose (*fast*) | 2.72* +3.0% | 2.79* +2.6% | 2.34* +2.2% | 4.19* +1.7% | 3.52* +3.2% |
| Propose (*slow*) | **2.64** | **2.72** | **2.29** | **4.12** | **3.41** |

*Evaluation.* Our method includes a comprehensive comparison between learning-based and non-learning-based algorithms. For non-learning-based algorithms, we selected PNG (*Boutell, 1997*), JPEG2000 (*Christopoulos, Skodras & Ebrahimi, 2000*), JPEG-LS (*Weinberger, Seroussi & Sapiro, 2000*), JPEG-XL (*Alakuijala et al., 2019*), FLIF (*Sneyers & Wuille, 2016*), and WebP (*WebEngines, 2000*). To ensure reproducibility, we chose learning-based methods with publicly available network model: L3C (*Mentzer et al., 2019*), L-Infinite (*Bai et al., 2021*), LC-FDNet (*Rhee et al., 2022*), LC-FDNet++ (*Rhee & Cho, 2023*), and DLPR (*Bai et al., 2024*).

## Performance comparisons

Table 2 presents the comparative results on the specified evaluation set. Evidently, our method, Propose (fast), outperforms both existing conventional codecs and learning-based codecs while Propose (slow) performs even better than Propose (fast). Compared to the traditional algorithm JPEG-XL, Propose (slow) shows an average improvement of 9.0%; compared to the learning-based algorithm DLPR, Propose (slow) averages a 4.5% improvement.

Furthermore, we found that in ultrasound and natural image datasets, our method, Propose (slow), significantly outperforms DLPR. However, in CT and X-ray datasets, the advantage of Propose (slow) is relatively smaller. This is because imaging methods like ultrasound produce images with more complex texture structures. The LC-L module used by Propose (slow) has more accurate inference capabilities for complex texture areas, hence showing superior performance in datasets with complex textures. In contrast, CT and X-ray imaging methods produce smoother images with fewer complex textures, which limits the scope for compression improvements by Propose (slow).

**Table 3 Summary of compression time comparisons.** Performance was measured in average encoding time per image (spp), the best performance is highlighted in bold, the second best performance is indicated with an asterisk (*).

| Method | Breast ultrasound images | Brain tumor MRI images | COVID-19 radiography | COVID-19 CT scan | COCO |
|---|---|---|---|---|---|
| L3C | 0.77 +83.3% | 0.27 +145.4% | 0.35 +191.7% | 0.69 +86.5% | 0.79 +132.3% |
| LC-FNet | 0.53* +26.2% | 0.13* +18.2% | 0.14* +16.7% | 0.41* +10.8% | 0.42* +23.5% |
| LC-FNet++ | 0.57 +35.7% | 0.13* +18.2% | 0.15 +25.0% | 0.44 +18.9% | 0.46 +35.3% |
| DLPR | 1.57 +273.8% | 0.49 +345.5% | 0.59 +391.7% | 1.18 +218.9% | 1.28 +276.5% |
| Propose (slow) | 0.65 +54.8% | 0.21 +90.9% | 0.22 +83.3% | 0.54 +45.9% | 0.56 +64.7% |
| Propose (fast) | **0.42** | **0.11** | **0.12** | **0.37** | **0.34** |

## Time comparisons

Table 3 displays the timing comparison results on the specified evaluation set, with experiments conducted on an Nvidia 2080Ti and Intel 12700. In tests across various types of datasets, Propose (fast) consistently demonstrated the fastest compression speed, highlighting its high practicality for real-world applications.

The speed advantage of Propose (fast) is attributed to the use of the LC-H module, which can obtain all residuals and residual probability distributions of subimages in a single inference. In contrast, Propose (slow) employs both LC-L and LC-H-Extend for more detailed subimages compression, resulting in Propose (slow) requiring twice the number of inferences as Propose (fast). Although Propose (slow) is slower than Propose (fast), it maintains a reasonable speed while offering state-of-the-art compression efficiency. Thus, Propose (slow) remains a highly efficient mode worth adopting.

The experiments did not include comparisons with traditional algorithms because traditional algorithms rely on CPU computations, whereas learning-based algorithms primarily depend on GPU computations, leading to a lack of comparability between the two. Additionally, the L-Infinite algorithm does not support GPU operations, to ensure fairness, it was excluded from the comparisons.

## Probability heatmap analysis

According to Shannon's information entropy theory, the optimal number of bits is $-\log_2 P(j)$, where $j$ is the residual value to be compressed, and $P(j)$ is the probability of $j$ predicted by the neural network. A higher $P(j)$ leads to better compression. In Fig. 7, different colors correspond to different values of the residual probability $P$. Red indicates $P$ is close to 0, meaning poor compression, while green indicates $P$ is close to 1, meaning excellent compression.

Figure 7A shows the probability heatmap of an ultrasound image. Due to the unique characteristics of ultrasound imaging, the texture is highly intricate. It can also be observed that red spots are widely distributed, indicating that the neural network often struggles with probability estimation in complex textured images. However, after applying the LC-H-Extend and LC-L modules, the number of red spots significantly decreases. This suggests that our Propose (slow) compression scheme can achieve more accurate probability

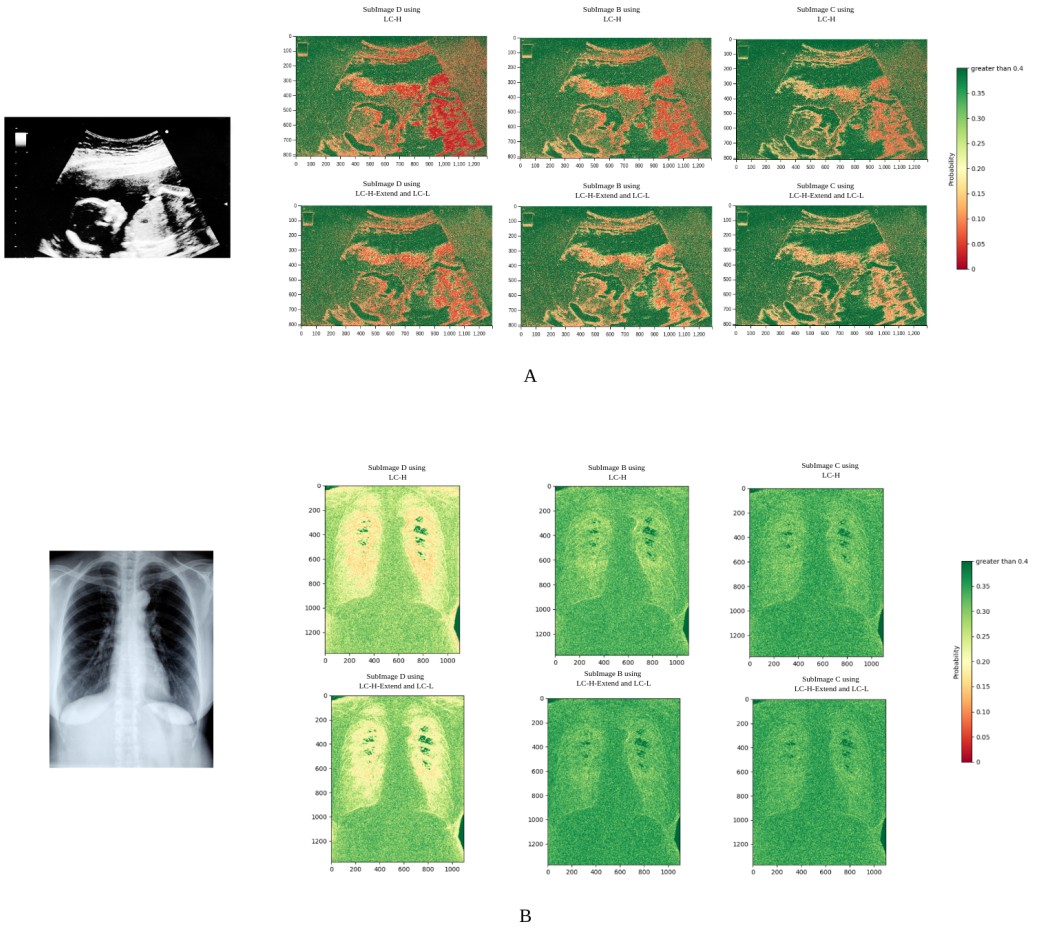

**Figure 7 Probability heatmaps.** In each set of subimages, the upper image group utilizes LC-H for probability estimation, while the lower image group employs LC-L and LC-H-Extend for probability estimation.

estimation in regions with complex textures, effectively addressing the issue of inaccurate probability estimation in these areas.

Figure 7B shows the probability heatmap of an X-ray image. Since the lungs are primarily filled with air, their texture is relatively simple. In Fig. 7B, the number of red spots is very low, regardless of whether LC-L or LC-H is used. This demonstrates that the neural network is highly efficient in reasoning when dealing with smooth images. After using the LC-H-Extend and LC-L combination for compression, the green areas become even more prominent, further proving that the Propose (slow) compression scheme can still enhance performance on smooth images.

Using the ultrasound image in Fig. 7A as an example, the experiment qualitatively analyzed residual probability values by dividing them into three ranges: greater than 0.1, between 0.01 and 0.1, and less than 0.01. Table 4 shows that after applying the LC-L and LC-H-Extend modules, the proportion of pixels with residual probability greater than 0.1 increases, while those below 0.01 decrease. This confirms that our Propose (slow)

**Table 4 Residual probability proportions.** This table presents the proportions of pixels within different residual probability ranges for subimages (D, B, C) using the LC-L and LC-H-Extend modules versus the LC-H module. The proportions are expressed as percentages.

| Module | Subimage | $P > 0.1$(%) | $0.01 \leq P \leq 0.1$(%) | $P \leq 0.01$ (%) |
|---|---|---|---|---|
| LC-L and LC-H-Extend | D | 64.24 | 30.09 | 5.67 |
| | B | 68.76 | 28.34 | 2.90 |
| | C | 70.58 | 26.64 | 2.78 |
| LC-H | D | 60.14 | 32.79 | 7.07 |
| | B | 61.89 | 31.80 | 6.31 |
| | C | 64.20 | 32.57 | 3.23 |

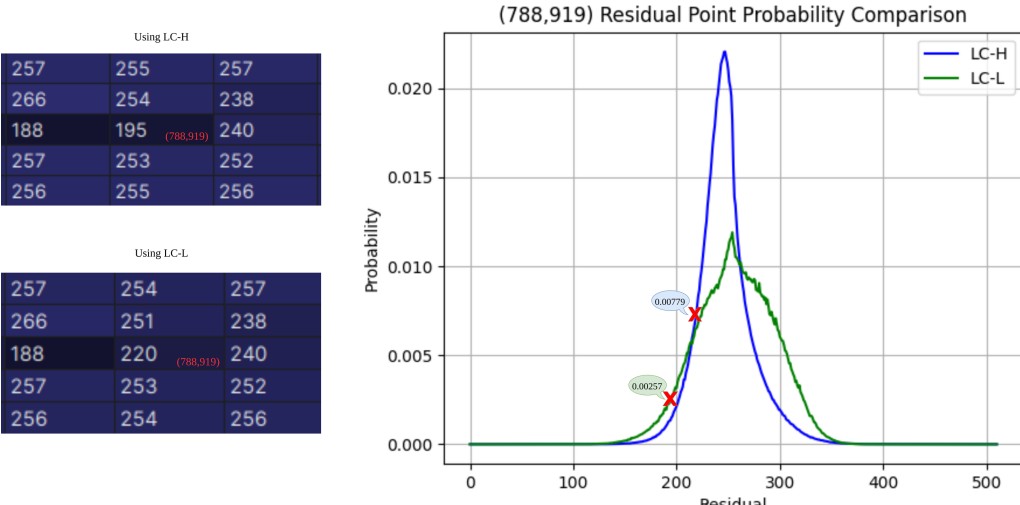

**Figure 8 Residual probability comparison.** The residual probability at coordinate (788,919) is estimated using both LC-H and LC-L.

compression scheme improves probability estimation, especially in complex textured regions.

## LC-L ablation experiment

The ablation experiment is analyzed from two perspectives: performance improvement for a single pixel and performance improvement for the entire subimage. As shown in Fig. 8, the experiment selects the residual point at the coordinates (788,919) as an example. The residual obtained by LC-H is 195, with a corresponding probability $P_{LC-H}(195) = 0.00257$. The residual obtained by LC-L is 220, with a corresponding probability $P_{LC-L}(220) = 0.00779$. According to Shannon's information entropy formula, bit $= -\log_2 P(j)$, the compression bit count for LC-H is 8.6 bits, and for LC-L, it is 7.4 bits. At (788,919), LC-L saves approximately 14% of the space compared to LC-H.

To evaluate the overall bits per pixel (bpp) improvement of LC-L on subimages (D, B, C), the experiment compares Model 1 using LC-H and Model 2 using LC-L. The network is evaluated on the Breast Ultrasound Images Dataset, excluding the JPEG-XL part (1.05 bpp).

**Table 5  Comparison of bpp improvement using LC-H and LC-L.** This table compares the bpp performance between Model 1 using LC-H and Model 2 using LC-L across different subimages (D, B, C) in the Breast Ultrasound Images Dataset.

| Subimage | Model 1 using LC-H | Model 2 using LC-L |
|---|---|---|
| D | 0.914 +8.6% | 0.841 |
| B | 0.388 +1.8% | 0.381 |
| C | 0.372 +0.5% | 0.370 |
| Total | 1.674 +5.1% | 1.592 |

Table 5 shows the ablation results for LC-L. Model 2 shows the greatest improvement on subimage D and the smallest improvement on subimage C. The reason for this difference lies in the input context of Model 1 during compression. When compressing subimage C, Model 1's input includes subimages A, D, and B, providing ample prior knowledge to accurately estimate probability in complex texture regions. However, when compressing subimage D, Model 1's input only includes subimage A, lacking sufficient prior knowledge, resulting in inaccurate probability inference. In this scenario, Model 2's input includes both subimages A and D-Left, with D-Left providing strong prior knowledge for D-Right, significantly enhancing probability estimation accuracy. Overall, using LC-L improved the model's performance by 5.1%.

# CONCLUSIONS

The primary academic contribution of this study lies in the proposal of a flexible tree-structured subimage segmentation mechanism, which significantly enhances the inference capabilities of neural networks in regions with complex textures by effectively leveraging subimage priors. Furthermore, the integration of an attention mechanism into the probabilistic estimation model extends its applicability in the domain of lossless compression. The results of the study indicate that the proposed flexible subimage segmentation scheme allows for the selective combination of subimages at different scales during the compression process, thereby enabling variable-speed compression. This approach achieves a balance between inference speed and accuracy, making it suitable for a variety of application scenarios and demonstrating broad applicability.

Despite the improvements in inference accuracy and compression flexibility achieved by this study, some limitations remain. Currently, the system requires loading both the LC-H and LC-L modules for variable-speed compression. Although these modules produce identical outputs, their differing input dimensions necessitate separate loading and processing, leading to high GPU memory usage and increased computational resource demands. This issue is particularly pronounced when inferring high-resolution image data. Additionally, while the relatively basic attention mechanism in the current network helps reduce computational cost and improve inference speed, it still has limitations in feature extraction. As a result, there is room for further enhancement in the model's inference capability.Lastly, the experiments in this study are designed for medical images, which are typically stored in a single-channel format. While the proposed method is effective for single-channel data, its applicability in multi-channel compression scenarios is limited.

Future research could focus on two main optimization strategies and extending the application domain. First, introducing a blank matrix of size $1 \times \frac{W}{2} \times \frac{H}{2}$ in the LC-H module would align its input dimensions with those of LC-L, allowing the system to load only the LC-H module. This adjustment would reduce memory consumption and computational redundancy. Second, incorporating more advanced attention mechanisms, as suggested by existing literature (*Ruan et al., 2022*; *Wang et al., 2023a*), from both channel and spatial perspectives could refine feature selection and enhance the network's inference capability and compression performance. These improvements would facilitate broader deployment of the system in resource-constrained environments while ensuring efficient and accurate compression in complex scenarios. For multi-channel compression, inter-channel correlations can be exploited by inferring multi-channel data from single-channel priors, thereby extending the application scope of this study.

### Funding
The authors received no funding for this work.

### Competing Interests
The authors declare there are no competing interests.

### Author Contributions
- Hengrui Liao conceived and designed the experiments, performed the experiments, analyzed the data, performed the computation work, prepared figures and/or tables, authored or reviewed drafts of the article, and approved the final draft.
- Yue Li conceived and designed the experiments, authored or reviewed drafts of the article, and approved the final draft.

### Data Availability
The LC-TMNet Source Code is available at Zenodo: Hengrui, L. (2024). LC-TMNet source code release. Zenodo. https://doi.org/10.5281/zenodo.11381738.

The LC-TMNet Model and Data are available at Zenodo: Hengrui, L. (2024). Dataset and Model in the LC-TMNet Paper. Zenodo. https://doi.org/10.5281/zenodo.11381358

This is a subset of the original dataset aimed at facilitating researchers in reproducing the conclusions of this experiment. The original full datasets are available at:

The Large COVID-19 CT Scan Slice Dataset is available at Kaggle:
https://www.kaggle.com/datasets/maedemaftouni/large-covid19-ct-slice-dataset.

The COVID-19 Radiography Dataset is available at Kaggle:
https://www.kaggle.com/datasets/preetviradiya/covid19-radiography-dataset.

The Brain Tumor Classification is available at Kaggle:
https://www.kaggle.com/datasets/sartajbhuvaji/brain-tumor-classification-mri.

The Breast Ultrasound Images Dataset is available at Kaggle:

https://www.kaggle.com/datasets/aryashah2k/breast-ultrasound-images-dataset.
The COCO dataset is available at: https://cocodataset.org/#download.

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
