# Peer review of "LC-TMNet: learned lossless medical image compression with tunable multi-scale network"

_PeerJ Computer Science, doi:10.7717/peerj-cs.2511_

## Round 0.1 · original submission · Major Revisions

· Academic Editor

Major Revisions

Dear authors,

Thank you for submitting your article. Reviewers have now commented on your article and suggest major revisions. We do encourage you to address the concerns and criticisms of the reviewers and resubmit your article once you have updated it accordingly. When submitting the revised version of your article, it will be better to address the following:

1. Pros and cons of the methods should be clarified. What are the limitation(s) methodology(ies) adopted in this work? Please indicate practical advantages, and discuss research limitations.
2. Space character should be correctly used.
3. Many of the equations are part of the related sentences. Attention is needed for correct sentence formation.
4. Equations should be used with correct equation number. Please do not use “as follows”, “given as”, etc. Explanation of the equations should also be checked. All variables should be written in italic as in the equations. Their definitions and boundaries should be defined. Necessary references should be provided.
5. All of the values for the parameters of all algorithms selected for comparison should be given.
6. Some more recommendations and conclusions should be discussed about the paper considering the experimental results. The conclusion section is weak. There is also no discussion section about the results. It should briefly describe the results of the study and some more directions for further research. You should describe the academic implications, main findings, shortcomings and directions for future research in the conclusion section. The conclusion in its current form is generally confused. What will be happen next? What we supposed to expect from the future papers? So rewrite it and consider the following comments:
- Highlight your analysis and reflect only the important points for the whole paper.
- Mention the benefits
- Mention the implication in the last of this section.

Best wishes,

Reviewer 1 ·

Basic reporting

The article titled “LC-TMNet: learned lossless medical image compression with tunable multi-scale network” provides a clear introduction, well-supported methodology and result analysis section. However,
1. For better understanding and readability, Figure 3 may be divided into two figures.
2. Better to provide the expanded form of abbreviations before its usage: SE attention, line 192

Experimental design

Proper citation is not done for
Christopoulos, C., Skodras, A., and Ebrahimi, T. (2000). The jpeg2000 still image coding system: an overview. IEEE transactions on consumer electronics, 46(4):1103–1127- line 356

Validity of the findings

No comment

Reviewer 2 ·

Basic reporting

This paper studies about medical image compression, and the contribution is mainly about Subimage Partitioning Scheme, attention mechanism, and the computational effecient design.

The writing should be improved. For example, be careful about formating. There should space after the colon ‘:’, stop '.', comma ',' and before parentheses '('.

Limited number of reference provided in the related work section.

There are too many subsection titles in each paragraph, which is unusual.

The attention mechanism should be discussed in detail with mathematical equations.

Experimental design

More baseline methods should be added.

In figure 8, I am curious why the background of 2 tables are not white?

Validity of the findings

The performance comparisons are fair enough.

Additional comments

N/A

·

Basic reporting

This paper tries to address the issue of inaccurate probability estimation by neural networks in complex texture regions. The proposed idea is very interesting, and the contribution is also presented properly and clearly. However, the following points should be taken care of in a possible revised version.
(1) First of all, the authors should describe the background of the paper and motivation of this paper in detail. In essence, there are many deep-learning based methods which employs the Unet or learned losses to promote the performance of segmentation or other vision tasks, authors should analyze their differences more clearly, and what is the novelty of your method.
(2) More technical details of your method should be given. I strongly suggest that authors should provide a Pseudocode Algorithm Diagram of your proposed method, which can make readers understand your methods more clearly.
(3) More visual results (qualitative analysis) should be included to enhance your conclusion.
(4) Conclusion should be enhanced in terms of future work.

Experimental design

no comment

Validity of the findings

no comment

Reviewer 4 ·

Basic reporting

1. The paper lacks an analysis of the disadvantages of current compression models.

2. The review primarily focuses on methodological aspects, with insufficient discussion on the motivation of compression on medical domain. More context on the specific topic is needed.

3. The authors appear to have overlooked some relevant literature, particularly learning-based methods for medical tasks.

Experimental design

no comment

Validity of the findings

no comment

Additional comments

no comment

Reviewer 5 ·

Basic reporting

All comments have been added in detail to the last section.

Experimental design

All comments have been added in detail to the last section.

Validity of the findings

All comments have been added in detail to the last section.

Additional comments

Review Report for PeerJ Computer Science
(LC-TMNet:learned lossless medical image compression with tunable multi-scale network)

1. Within the scope of the study, a new lossless image compression method is proposed, which increases the accuracy of probabilistic estimation of uncoded subimages and uses an image segmentation mechanism.

2. In the introduction, the studies on learning-based lossless compression algorithms in the literature, advancement of imaging technology, recent studies, subimage partitioning scheme with a tree structure and the main contributions of the study are clearly mentioned at a sufficient level.

3. Although the Related Works section is discussed in terms of both Multi-Scale Entropy and Likelihood-Based Generative models, this section definitely needs to be detailed. Here, it is recommended to add a literature table or related explanations that consist of important parts such as advantages, disadvantages, results etc. of the studies in the literature regarding the models considered from both perspectives and emphasize the main differences and positive aspects of this study from the literature.

4. Within the scope of the proposed method in the study, implementation of LC-L, LC-H and LC-H-Extend architectures, overview of the compression process, implementation of attention-enhanced Unet architecture are sufficiently mentioned and the originality point of the study is clearly stated. Here, it should be explained how the preprocessing operations in LC-H and LC-L architectures are determined and how the parameters are determined with the reasons for the preference of loss functions and whether different experiments have been done.

5. Dataset selection is very important to increase the usability of the proposed model and to prove it. In this context, it is positive to choose various medical image datasets. However, although there are many open source datasets in the literature, please explain in detail why these are preferred and/or the reasons for choosing different datasets.

6. Although the models selected for compression performance comparison are at a certain level in terms of number and variety, it is recommended to compare the results with a few state-of-the-art models in order to better emphasize the superiority of the proposed model.

7. The results obtained and the conclusion section emphasize the importance of the study, and it is recommended that detailed information be included in the last section, especially in terms of future works.

As a result, although the study is very interesting in terms of subject and usability and has the potential to make a significant contribution to the literature, all the sections listed above should be read very carefully, each of them should be answered in detail, and the relevant changes should be made completely in the relevant section of the paper.

---

## Round 0.2 · Minor Revisions

· Academic Editor

Minor Revisions

Dear authors,

Thank you for your paper. According to one reviewer, your paper still needs revision and we encourage you to address the concerns and criticisms of Reviewer 4 and resubmit your article once you have updated it accordingly.

Best wishes,

Reviewer 2 ·

Basic reporting

The paper titled "LC-TMNet: Learned Lossless Medical Image Compression with Tunable Multi-Scale Network" by Hengrui Liao and Yue Li presents a novel approach to lossless medical image compression, a critical task in medical imaging where precise image representation is essential for accurate clinical diagnosis. The authors propose a method that leverages a flexible tree-structured image segmentation mechanism to improve the accuracy of neural networks in estimating probabilities for complex textured regions, which traditional methods struggle with.

Author have provided a detailed point by point response.

Experimental design

All underlying data used in this study have been provided, ensuring transparency and allowing for robust, statistically sound analysis.
The data and methods are well-controlled, offering a solid foundation for both the original findings and any subsequent replication efforts.

The experimental design is comprehensive, and Table 2,3,4,5, and Figure 7 are well designed.

Validity of the findings

The impact of this research lies in its potential to significantly enhance the efficiency and accuracy of medical image compression, a critical area in clinical diagnostics where lossless compression is essential.
The novelty is evident in the combination of tree-structured segmentation and attention mechanisms, which together improve the handling of complex textures in medical images, a challenge that previous methods have not adequately addressed.

Additional comments

1 The english writing is fine.

2 It is uncommon in Figure 8, that the background colour of table is in dark blue. (Looks like a screenshot, rather than a table.)

·

Basic reporting

This study proposes a new lossless image compression method that employs a ûexible tree-structured image segmentation mechanism.

Experimental design

Experimental design is reasonable.

Validity of the findings

Validity of the findings is comprehensive.

Additional comments

The authors have addressed all of my concerns, now this version can be accepted for publication in this reputated journal.

Reviewer 4 ·

Basic reporting

Thank the authors for the classification. But I still have some concerns.

1. The limitations of the methods is not clearly stated.
2. The motivation of the proposed method is unclear in medical imaging. Since the clinical trail require the quality of medical images.

Experimental design

The experimental design is unfair.

Validity of the findings

Thank the authors for the classification. But I still have some concerns.

1. The limitations of the methods is not clearly stated.
2. The motivation of the proposed method is unclear in medical imaging. Since the clinical trail require the quality of medical images.
3. The experimental design is unfair.

Reviewer 5 ·

Basic reporting

All comments have been added in detail to the last section.

Experimental design

All comments have been added in detail to the last section.

Validity of the findings

All comments have been added in detail to the last section.

Additional comments

Thank you for the revision. It is observed that the responses to the reviewer comments and the changes made to the paper are sufficient. For this reason, I recommend that the paper be accepted.

---

## Round 0.3 · Minor Revisions

· Academic Editor

Minor Revisions

Dear authors,

Thank you for the revised paper. Although your paper seems to be improved, please make some edits to address the concerns of Reviewer 4 to highlight the motivation and novelty of the proposed method with considering the clinical specialties.

Best wishes,

Reviewer 4 ·

Basic reporting

Thank the authors for the classification. But it still have the following concerns. 1) the motivation of the task, and 2) limited novelty of the proposed method without considering the clinical specialties.

Experimental design

it is ok

Validity of the findings

it is ok

---

## Round 0.4 · accepted · Accept

· Academic Editor

Accept

Dear authors,

Thank you for the third revision. Four of the origianl reviewers think that your paper is improved and can be accepted for publication. Although one of the reviewer has some concerns, I also think that the paper is sufficiently improved and seems acceptable for publication.

Best wishes,

Reviewer 1 ·

Basic reporting

The authors have addressed my review comments in the resubmission.

Experimental design

The two experimental modes, slow mode and fast mode, have been carefully balanced to evaluate the performance in terms of compression ratio and speed.

Validity of the findings

The novelty of the proposed architecture is evident from the benchmarking provided in the manuscript.

Reviewer 4 ·

Basic reporting

The proposed work’s limited motivation makes it challenging to adopt in real-world clinical environments.

Experimental design

The comparison is unfair.

Validity of the findings

n/a